# Nationwide Population-Based Epidemiological Study for Outcomes of Adjunctive Steroid Therapy in Pediatric Patients with Bacterial Meningitis in Taiwan

**DOI:** 10.3390/ijerph18126386

**Published:** 2021-06-12

**Authors:** Dong-Yi Hsieh, Yun-Ru Lai, Chia-Yi Lien, Wen-Neng Chang, Chih-Cheng Huang, Ben-Chung Cheng, Chia-Te Kung, Cheng-Hsien Lu

**Affiliations:** 1Department of Neurology, Kaohsiung Chang Gung Memorial Hospital, College of Medicine, Chang Gung University, Kaohsiung 833, Taiwan; b9202095@cgmh.org.tw (D.-Y.H.); yunrulai@cgmh.org.tw (Y.-R.L.); u9301024@cgmh.org.tw (C.-Y.L.); cwenneng@ms19.hinet.net (W.-N.C.); hjc2828@gmail.com (C.-C.H.); 2Department of Medicine, Kaohsiung Chang Gung Memorial Hospital, College of Medicine, Chang Gung University, Kaohsiung 833, Taiwan; benzmcl@gmail.com; 3Department of Emergency Medicine, Kaohsiung Chang Gung Memorial Hospital, College of Medicine, Chang Gung University, Kaohsiung 833, Taiwan; kungchiate@gmail.com; 4Center for Shockwave Medicine and Tissue Engineering, Kaohsiung Chang Gung Memorial Hospital, College of Medicine, Chang Gung University, Kaohsiung 833, Taiwan; 5Department of Biological Science, National Sun Yat-Sen University, Kaohsiung 804, Taiwan; 6Department of Neurology, Xiamen Chang Gung Memorial Hospital, Xiamen 361000, China

**Keywords:** adjunctive steroid therapy, pediatric bacterial meningitis, nationwide population-based epidemiological study, Taiwan

## Abstract

Although corticosteroids can serve as an effective anti-inflammatory adjuvant therapy, the role of adjunctive steroid therapy in pediatric bacterial meningitis in Taiwan remains under-investigated. Cases of acute bacterial meningitis, aged between 1 month and 20 years, were divided into a steroid group (empirical antibiotics with adjunctive steroid therapy) and a non-steroid group (empirical antibiotics only). Data were identified from the annual hospitalization discharge claims of the National Health Insurance Research Database using the International Classification of Diseases, Ninth Revision codes. Of the 8083 episodes enrolled in this study, 26% (2122/8083) and 74% (5961/8083) were divided into the steroid and non-steroid groups, respectively. The fatality rates were 7.9% in the steroid group and 1.7% in the non-steroid group during hospitalization (*p* < 0.0001). In the steroid and non-steroid groups, the median length of hospital stay was 13 and 6 days, respectively (*p* < 0.0001). Medical costs (median (interquartile range)) of hospitalization were 77,941 (26,647–237,540) and 26,653 (14,287–53,421) New Taiwan dollars in the steroid and non-steroid groups, respectively (*p* < 0.0001). The steroid group had a more fulminant course at baseline, a higher fatality rate, length of hospital stay, and medical cost of hospitalization. Therefore, the beneficial effects of the adjunctive use of corticosteroids in pediatric bacterial meningitis are inconclusive, and additional prospective multicenter investigations are required to clarify this issue.

## 1. Introduction

Bacterial meningitis in pediatric patients is a medical emergency that results from bacterial infection of the leptomeninges surrounding the brain and spinal cord. Despite appropriate antimicrobial therapy and advanced neurocritical care, pediatric bacterial meningitis carries a high mortality rate, and neurological sequelae are common among survivors [1,2,3,4]. Rapid initiation of empirical antibiotics after a lumbar puncture for highly suspected cases of bacterial meningitis is recommended by evidence-based medicine. However, there is insufficient medical evidence for the recommendation of adjunctive corticosteroids.

Several studies have demonstrated that children with acute bacterial meningitis treated with adjunctive corticosteroid therapy had a significant reduction in hearing impairment and neurological sequelae. However, they had similar results in overall mortality rate and length of hospital stay [2,5,6]. In clinical practice, certain concerns have been raised regarding the initiation of adjuvant corticosteroid therapy for pediatric bacterial meningitis. This is due to the limited evidence and conflicting results. The efficacy of corticosteroid therapy in reducing hearing loss has been demonstrated in children with *Haemophilus influenzae* type b (Hib) meningitis but not in children with meningitis due to non-*Haemophilus* species [5]. The timing of administering dexamethasone was suggested prior to or at the same time as the first dose of antimicrobial therapy, and might be less beneficial if administered later [5,7,8,9]. The American Academy of Pediatrics recommends vancomycin plus either cefotaxime or ceftriaxone as empirical antimicrobial therapy for suspected bacterial meningitis in children aged 1 month and above [10]. However, prior studies have demonstrated that the adjunctive use of dexamethasone significantly reduced the penetration of vancomycin into the cerebral spinal fluid (CSF) and bactericidal activity in experimental penicillin- and cephalosporin-resistant pneumococcal meningitis [11,12,13].

Depending on the causative organism, severity at the time of admission, and underlying condition of the patient, the precise recommendations for adjuvant corticosteroid therapy have not been clearly defined. Therefore, this study was designed to report the bacteriology, clinical features, and outcomes of adjuvant corticosteroid therapy in pediatric bacterial meningitis in Taiwan.

## 2. Patients and Methods

### 2.1. Data Source

The National Health Insurance (NHI) service was launched in Taiwan on 1 March 1995, and provides nationwide comprehensive medical coverage for the entire population. In 2003, the National Health Insurance Bureau completed a comprehensive integrated circuit (IC) card application through the National Health Insurance Card organization. The card holder’s medical records were integrated onto the computer chip of an individual person’s card and uploaded onto a cloud server during each seeking of medical care. The National Health Insurance Agency exported the health insurance data from the previous year and selected files in the middle of each year for research purposes. After encryption of the identity field, the data were submitted to the NHI to augment the National Health Insurance Research Database (NHIRD). The current national health insurance service coverage rate is over 99%. This designates the NHIRD as a representative, empirical, and significant digital database for the fields of medical and health-related research. The NHIRD is an important research resource and is accessible to scientists in Taiwan, and due to encrypted property, the information of individuals or medical institutions at any level cannot be obtained by the researcher.

### 2.2. Study Subjects

In this nationwide population-based epidemiological study, we collected the data of children aged between 1 month and 20 years who were treated for bacterial meningitis between 1 January 2000 and 31 December 2013 (Figure 1). The patients’ baseline characteristics were recorded in hospitalization discharge claims, including an encrypted personal identification number, date of birth, length of stay, relevant department, costs, and the first five International Classification of Diseases, Ninth Revision (ICD-9) codes for bacterial meningitis (i.e., 003.21, 027.0, 036.0, 320.0, 320.1, 320.2, 320.3, 320.7, 320.81, 320.82, 320.89, and 320.9). Patients with tuberculous meningitis were excluded from this study.

There were 7561 cases of bacterial meningitis, of which 522 cases represented recurrent episodes. In total, there were 8083 episodes of bacterial meningitis during the study period. Of these, 26.3% (2122/8083) and 73.7% (5961/8083) were placed in the steroid and non-steroid groups, respectively. We compared empirical antibiotic treatment in combination with systemic corticosteroids (steroid group) with empirical antibiotic treatment alone (non-steroid group) in patients with acute bacterial meningitis. The selection of empirical antimicrobial regimens followed the recommended guidelines for pediatric bacterial meningitis. The recommended antimicrobials were prescribed to cover the likely pathogens in the affected age group.

### 2.3. Outcome Assessment

Data are expressed as the mean ± standard deviation or median (interquartile range). Categorical variables were compared using the Chi-square test or Fisher’s exact test. Demographic data including sex, clinical features, duration, and cost of hospitalization, as well as outcomes between the steroid and non-steroid groups, were compared. Age at infection is expressed as mean ± standard deviation. Data of length of hospital stay (days) and medical costs of hospitalization (New Taiwan dollars, NTD) are expressed as median (interquartile range (IQR)). The effects of individual variables on in-hospital and 1-year fatality rates, including sex, age, and study groups (in the steroid and non-steroid groups), were analyzed using the univariate Cox proportional hazards model. Any significant univariate Cox proportional hazards model was repeated with adjustments based on a multivariate Cox proportional hazards model. Finally, the association of the in-hospital and 1-year fatality rates with the survival curve between the two patient groups (steroid and non-steroid groups) was assessed using Kaplan–Meier plots and compared using the log-rank test (Figure 2). All statistical analyses were conducted using the SAS software package, version 9.1 (2002, SAS Statistical Institute, Cary, NC, USA).

## 3. Results

### 3.1. Baseline Characteristics of Study Patients

The enrollment of patients is shown in Figure 1 and the baseline patient characteristics are listed in Table 1. The 7561 cases included 4573 male (60.5%) and 2988 female (39.5%) patients. The causative pathogens isolated from CSF cultures of the 8083 episodes of bacterial meningitis are shown in Table 2. *Streptococcus (S.) pneumoniae* was the most frequent causative pathogen, accounting for 7.2% and 1.9% of episodes in the steroid group and non-steroid group, respectively, followed by *Haemophilus (H.) influenzae* (4.0% and 0.6%) and *Escherichia (E.) coli* (0.7% and 0.3%).

### 3.2. Underlying Diseases and Clinical Features

Chronic epilepsy, hypertension, type 1 diabetes mellitus, and status post neurosurgical procedures were the most common underlying conditions in both the steroid and non-steroid groups (Table 3). In the steroid group, hydrocephalus (11.0%) was the most common clinical feature, followed by acute respiratory failure (10.2%) and pneumonia (9.3%). In the non-steroid group, urinary tract infection (10.6%) was the most common clinical feature, followed by fever (6.7%) and pneumonia (4.0%).

### 3.3. Duration and Cost of Hospitalization

The median length of hospital stay (days) was 13 (range: 6–27) and 6 (range: 4–10) days in the steroid and non-steroid groups, respectively (*p* < 0.0001), (Table 4). The medical cost of hospitalization (New Taiwan dollars (NTD), median (IQR)) was NTD 77,941 (26,647–237,540) and 26,653 (14,287–53,421) in the steroid and non-steroid groups, respectively (*p* < 0.0001).

### 3.4. Therapeutic Outcome of Patients 

The fatality rates were 7.9% (167/2122) in the steroid group and 1.7% (100/5961) in the non-steroid group during hospitalization (*p* < 0.0001). The fatality rates were 13.7% (290/2122) and 3.3% (198/5961) in the steroid and non-steroid groups, respectively, during the 1-year follow-up period (*p* < 0.0001). Furthermore, the hazard ratios (relative to the non-steroid group) of the in-hospital and 1-year fatality rates were 2.6 (95% CI: 2.0–3.3; *p* < 0.0001) and 4.13 (95% CI: 3.4–5.0; *p* < 0.0001) in the multivariate Cox model after adjustment for both age and sex, respectively (Table 5 and Table 6). To look at each factor individually, we calculated Kaplan–Meier estimates of the fraction, developing the survival probability (survival and non-survival outcome) in patients with acute bacterial meningitis between two groups (steroid and non-steroid groups) during the 1-year follow-up period and tested for differences by using a log-rank test. The results showed that patients in the steroid group had a statistically significant worse survival outcome (*p* < 0.0001) (Figure 2).

## 4. Discussion

Acute bacterial meningitis is a life-threatening inflammatory response in the meninges and subarachnoid space caused by bacterial infection. Considering the anti-inflammatory potential and production modulation of cytokines, systemic corticosteroids have been alleged to prevent neurologic complications or even reduce the mortality rate of bacterial meningitis. One retrospective cohort study of 2780 children with bacterial meningitis in the United States between 2001 and 2006 demonstrated an overall mortality rate of 4.2%. Adjuvant corticosteroids did not reduce mortality (6.0% and 4.0% in the steroid and control groups, respectively) [2]. A meta-analysis of the use of corticosteroids for acute bacterial meningitis showed no significant reduction in overall mortality, but demonstrated a reduction in mortality rate for *S. pneumoniae* meningitis in subgroup analyses for causative organisms [5]. One recent meta-analysis investigating the role of adjunctive dexamethasone therapy in pediatric bacterial meningitis demonstrated that dexamethasone had no significant effect on the follow-up mortality [6]. Our results showed that the mortality rate was significantly higher in the steroid group than in the non-steroid group, regardless of in-hospital fatality or 1-year fatality. We speculate that clinicians are more likely to administer systemic steroids to patients who are in a more critical state. This has been previously reported to be beneficial to patients with sepsis [14]. Due to limited data, this study did not perform subgroup analyses of reduction in mortality for causative organisms.

In our study, females had significantly higher in-hospital fatality and 1-year fatality. Sex differences in health and mortality have been of longstanding interest to researchers, and several studies have been devoted to this topic. One prospective analysis investigating sex difference in adults with community-acquired bacterial meningitis showed that male sex was an independent predictor of unfavorable outcome and death [15]. In contrast, another retrospective descriptive study reported females with urgent treatable etiologies to be a predictive factor of poor outcome [16]. For pediatric bacterial meningitis, the studies that explored sex-based differences in mortality are scarce and have had inconsistent results [17,18]. The reason for discrepancy in mortality between males and females was uncertain; however, it might be due to the differences of underlying development processes, and immune and endocrine systems [19].

The causative pathogens of bacterial meningitis in children vary widely according to age, geographic region, or the patient’s underlying medical condition [20,21]. Although the overall incidence dropped after the introduction of widespread pneumococcal vaccination, *S. pneumoniae* remains the most common cause of pediatric bacterial meningitis [22,23]. One multi-state surveillance study in the United States in 1986 reported that *H. influenzae* was the most common cause of bacterial meningitis in children prior to widespread vaccination [24]. A 29-year-long assessment of pediatric bacterial meningitis in northern Taiwan between 1984 and 2012 demonstrated that Group B *Streptococcus* and *S. pneumoniae* were the most common causative pathogens in the early period of the study. However, *E. coli* meningitis increased and became the most common pathogen in the later period. In our study, *S. pneumoniae* was the most frequently identified pathogen, whereas *H. influenzae* accounted for 4.01% and 0.51% of episodes in the steroid and non-steroid groups, respectively. The introduction of the Hib vaccine in Taiwan in 1993 played an important role in reducing the incidence of Hib infection. This might contribute to the relatively low incidence of *H. influenzae* meningitis [25,26]. In Taiwan, a pneumococcal vaccination program for children aged 2 to 5 years was initiated in 2013. Further surveillance data to identify its effectiveness in the prevention of *S. pneumoniae* meningitis are warranted.

In this study, there was a decreasing trend in the numbers of pediatric bacterial meningitis during the study period, and a similar trend was also reported in prior study [25]. We presumed that it was attributed to numerous factors, including lower birth rate, better environmental sanitation, and vaccination administration.

Despite appropriate antimicrobial treatment, neurologic complications are frequently encountered among childhood survivors of bacterial meningitis [1,27,28,29,30]. The efficacy of corticosteroid therapy in reducing hearing loss has been observed in children with Hib meningitis, but this efficacy was not demonstrated in children with meningitis caused by non-*Haemophilus* species [5,8]. One recent meta-analysis also showed that adjunctive administration of dexamethasone decreased the possibility of hearing loss and severe neurological sequelae [6]. In this study, the most common neurologic complication in the steroid group was hydrocephalus, followed by brain edema and brain abscess. In the non-steroid group, hydrocephalus and brain edema were also the most common neurologic complications. However, the proportions of both were significantly lower than those in the steroid group. In another retrospective observational study in Taiwan, regardless of the steroid or non-steroid group, brain abscess was identified as the most common neurologic complication in adult bacterial meningitis. However, only a small portion of patients presented with hydrocephalus and brain edema [31]. Our study also showed a significantly higher incidence of respiratory failure and pneumonia in the steroid group. This might indicate that clinicians tended to administer systemic corticosteroids to severely ill patients.

Although we had a large sample size to perform an overall analysis from the NHIRD of Taiwan, our study has some limitations. First, several previous studies have demonstrated that the annual incidence of bacterial meningitis in children and causative pathogens varied with age [20,21,22,25]. Except for *H. influenzae* (ICD-9-CM codes 3200), *S. pneumoniae* (ICD-9-CM codes 3201), and *Escherichia coli* (ICD-9-CM codes 320.82), not all of the causative pathogens had ICD-9-CM codes that could be recorded in the NHIRD nationwide population-based database. Therefore, we could not analyze the outcomes of adjunctive steroid therapy for certain causative pathogens. Consequently, the role of the adjunctive use of corticosteroids in pediatric cases with acute bacterial meningitis involving a single pathogen remains unclear. Second, clinicians were prone to prescribe adjunctive steroid therapy in pediatric patients who had a more fulminant clinical course (e.g., severe sepsis, respiratory failure, or cerebral infarctions superimposed on bacterial meningitis). These patients inevitably had unfavorable outcomes. In addition, the timing of administering dexamethasone was suggested prior to, or at the same time as, the first dose of antimicrobial therapy and might be less beneficial if administered later. Due to the nature of database analysis, our data did not include the timing between administration of corticosteroids and antibiotics. Third, the patients in the two groups might have had different disease severity or an underlying disease. Conducting a matched analysis based on severity or a different underlying disease of the two groups would make the results more convincing, and further studies might be needed to clarify the influence of the underlying disease on the outcome. Finally, it was important to explore the improvement of long-term sequelae with adjunctive corticosteroid therapy, especially in certain causative pathogens, but this study did not include such analysis due to limited data. It would be useful to have more data on the clinical manifestation of each patient so as to better evaluate neurologic outcome after adjunctive corticosteroid therapy. Therefore, the role of adjunctive corticosteroid therapy was hard to evaluate, and a statistical bias existed in the study.

The steroid group had more fulminant courses (e.g., hydrocephalus, acute respiratory failure, pneumonia, brain edema, brain abscess, and acute symptomatic seizure) at baseline. It was inevitable that this group had a higher fatality rate, length of hospital stay, and medical cost of hospitalization. The beneficial effects of adjunctive corticosteroid therapy on pediatric bacterial meningitis remain inconclusive. Additional prospective multicenter investigations are warranted to clarify the role of adjunctive corticosteroid therapy in pediatric bacterial meningitis in Taiwan.

## Figures and Tables

**Figure 1 ijerph-18-06386-f001:**
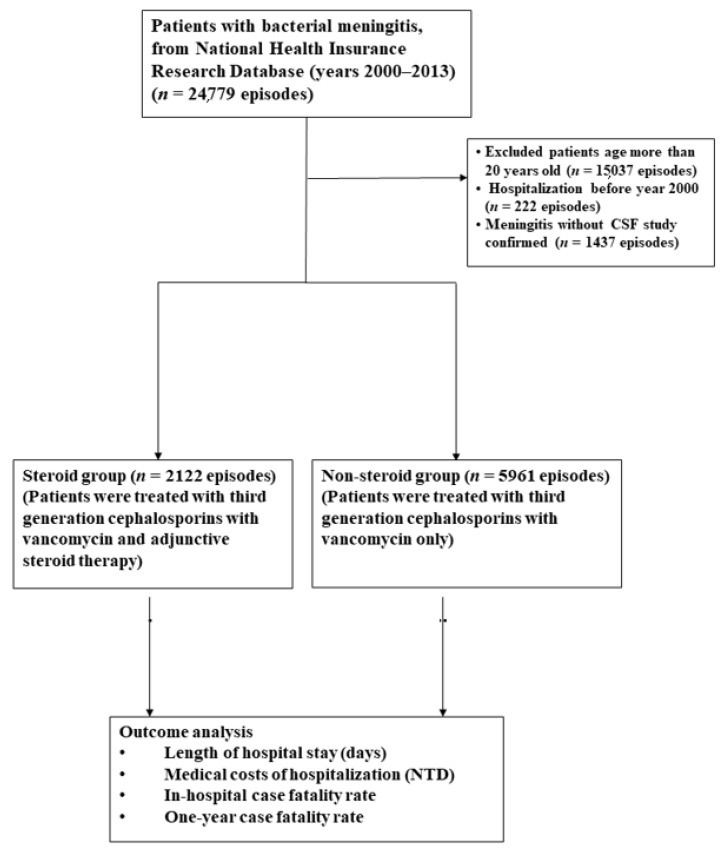
Enrollment of pediatric patients.

**Figure 2 ijerph-18-06386-f002:**
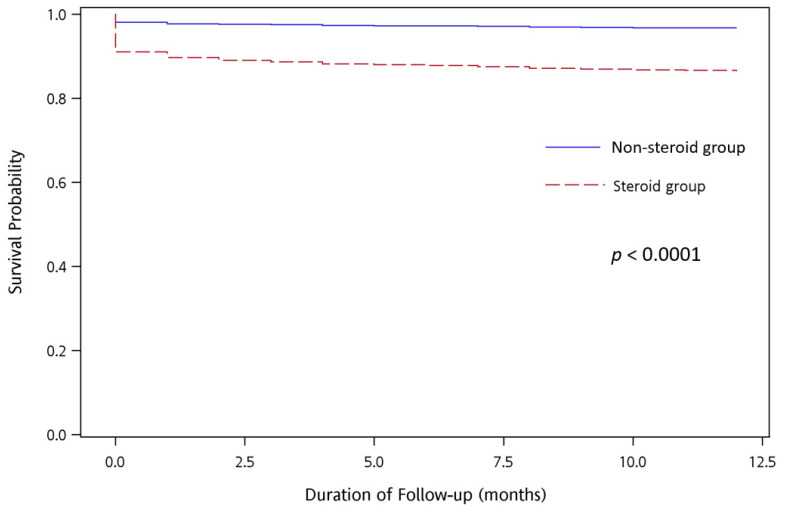
Kaplan–Meier plots indicating the percentage of survival probability in pediatric patients with acute bacterial meningitis during the 1-year follow-up period (*p*-value was obtained by log-rank comparison of data).

**Table 1 ijerph-18-06386-t001:** Baseline characteristics of patients.

Patient Enrollment	Steroid Group (*n* = 2122)	Non-Steroid Group (*n* = 5961)	*p*-Value
	*n* (%)	*n* (%)	
Age at infection (mean ± SD, years)	5.5 ± 4.0	4.4 ± 3.6	
Sex			0.63
Male	1275(60.08%)	3617 (60.68%)	
Female	847 (39.92%)	2344(39.32%)	
Distribution of years ^§^			<0.0001
2000	313 (14.75%)	724 (12.15%)	
2001	353 (16.64%)	1254 (21.04%)	
2002	227 (10.70%)	668 (11.21%)	
2003	159 (7.49%)	393 (6.59%)	
2004	163 (7.68%)	390 (6.54%)	
2005	162 (7.63%)	540 (9.06%)	
2006	151 (7.12%)	409 (6.86%)	
2007	93 (4.38%)	333 (5.59%)	
2008	95 (4.48%)	242 (4.06%)	
2009	84 (3.96%)	231 (3.88%)	
2010	93 (4.38%)	186 (3.12%)	
2011	93 (4.38%)	207 (3.47%)	
2012	76 (3.58%)	194 (3.25%)	
2013	60 (2.83%)	190 (3.19%)	

^§^ = There were 7561 cases of bacterial meningitis. Of these, 522 had recurrent episodes. In total, 8083 episodes suffered from bacterial meningitis during the study period. SD: standard deviation.

**Table 2 ijerph-18-06386-t002:** Causative pathogens.

Causative Pathogens ^§^	Steroid Group (*n* = 2122) *n* (%)	Non-Steroid Group (*n* = 5961) *n* (%)
Streptococcus Species (*n* = 569)		
*Streptococcus Pneumoniae*	152 (7.2%)	110 (1.9%)
Other streptococci	127 (6.0%)	180 (3.0%)
***Haemophilus influenzae*** **(** ***n*** **= 119)**	85 (4.0%)	34 (0.6%)
Gram-negative bacilli (*n* = 175)		
*Escherichia coli*	14 (0.7%)	18 (0.3%)
Pseudomonas species	2 (0.09%)	5 (0.08%)
Salmonella species	3 (0.1%)	1 (0.02%)
Other Gram-negative Bacilli	54 (2.5%)	78 (1.3%)
Staphylococcus Species (*n* = 20)		
*Staphylococcus aureus*	10 (0.5%)	9 (0.2%)
Other Staphylococci	1 (0.05%)	0
***Neisseria meningitidis*** **(*n* = 1)**	0	1 (0.02%)
***Listeria monocytogenes*** **(*n* = 10)**	7 (0.3%)	3 (0.05%)
**Anaerobes pathogens (*n* = 10)**	7 (0.3%)	3 (0.05%)
Others (*n* = 7179)	1660 (78.2%)	5519 (92.6%)

^§^ = There were 7561 cases of bacterial meningitis. Of these, 522 had recurrent episodes. In total, 8083 episodes suffered from bacterial meningitis during the study period. *Haemophilus influenzae* (ICD-9-CM codes 3200), *Streptococcus pneumoniae* (ICD-9-CM codes 3201), *Escherichia coli* (ICD-9-CM codes 320.82), ICD-9-CM = International Classification of Diseases.

**Table 3 ijerph-18-06386-t003:** Underlying diseases and clinical features.

Characteristics	Steroid Group (*n* = 2122)	Non-Steroid Group (*n* = 5961)	*p*-Value
	*n* (%)	*n* (%)	
Underlying diseases			
Chronic epilepsy	45 (2.1%)	116 (2.0%)	0.62
Hypertension	46 (2.2%)	28 (0.5%)	<0.0001
Type 1 diabetes mellitus	40 (1.9%)	23 (0.4%)	<0.0001
Status post neurosurgical procedure	25 (1.2%)	26 (0.4%)	0.0002
Systemic lupus erythematosus	14 (0.7%)	3 (0.05%)	<0.0001
Chronic kidney diseases	8 (0.4%)	10 (0.2%)	0.08
Non-alcoholic liver cirrhosis	6 (0.3%)	6 (0.1%)	0.06
Atrial fibrillation	6 (0.3%)	4 (0.07%)	0.0152
Clinical features			
Hydrocephalus	234 (11.0%)	157 (2.6%)	<0.0001
Acute respiratory failure	216 (10.2%)	67 (1.1%)	<0.0001
Pneumonia	198 (9.3%)	240 (4.0%)	<0.0001
Urinary tract infection	184 (8.7%)	630 (10.6%)	0.0126
Brain edema	90 (4.2%)	71 (1.2%)	<0.0001
Brain abscesses	84 (4.0%)	46 (0.8%)	<0.0001
Fever	73 (3.4%)	398 (6.7%)	<0.0001
Headache	40 (1.9%)	205 (3.4%)	0.0003
Shunt infection	38 (1.8%)	61 (1.0%)	0.0058
Acute symptomatic seizure	30 (1.4%)	40 (0.7%)	0.0015
CSF rhinorrhea	17 (0.8%)	21 (0.4%)	0.0094
Bacterial endocarditis	6 (0.3%)	2 (0.03%)	0.0017
Cerebral infarctions	4 (0.2%)	9 (0.2%)	0.71

**Table 4 ijerph-18-06386-t004:** Duration and cost of hospitalization.

Hospitalization	Steroid Group	Non-Steroid Group	*p*-Value
Length of hospital stay (days) (Median (IQR))	13 (6–27)	6 (4–10)	*p* < 0.0001
Medical costs of hospitalization (NTD) (Median (IQR))	77,941 (26,647–237,540)	26,653 (14,287–53,421)	*p* < 0.0001

Median (IQR); IQR = interquartile range; TWD: New Taiwan dollars.

**Table 5 ijerph-18-06386-t005:** In-hospital fatality.

Fatality	Survival *n* (%)	Death *n* (%)	Univariate Cox Model		Multivariate Cox Model	
			Hazard Ratio ^§^ (95% CI)	*p*-Value	Hazard Ratio ^§^ (95% CI)	*p*-Value
Study groups						
Non-steroid group	5861 (98.3%)	100 (1.7%)	2.531 (1.9–3.3)	<0.0001	2.551 (2.0–3.3)	<0.0001
Steroid group	1955 (92.1%)	167 (7.9%)				
Age at infection (mean, years)	4.6	5.3	0.996 (1.0–1.1)	0.75	0.992 (0.9–1.0)	0.50
Sex						
Male	4754 (97.2%)	138 (2.8%)	1.441 (1.1–1.8)	0.0029	1.436(1.1–1.8)	0.0032
Females	3062 (96.0%)	129 (4.0%)				

^§^ = Relative to non-steroid group.

**Table 6 ijerph-18-06386-t006:** One-year fatality.

Fatality	Survival *n* (%)	Death *n* (%)	Univariate Cox Model		Multivariate Cox Model	
			Hazard Ratio ^§^ (95% CI)	*p*-Value	Hazard Ratio ^§^ (95% CI)	*p*-Value
Study groups						
Non-steroid group	5763 (96.7%)	198 (3.3%)	4.26 (3.56–5.1)	<0.0001	4.13 (3.443–4.953)	<0.0001
Steroid group	1832 (86.3%)	290 (13.7%)				
Age at infection (mean, years)	4.6	5.7	1.04 (1.02–1.06)	<0.0001	1.026 (1.009–1.042)	0.0021
Sex						
Male	4622 (94.5%)	270 (5.5%)	1.25 (1.04–1.49)	0.016	1.255 (1.05–1.501)	0.0127
Females	2973 (93.2%)	218 (6.8%)				

^§^ = Relative to non-steroid group.

## Data Availability

The data presented in this study are available on request from the corresponding author.

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
