# Peer review of "Nationwide Population-Based Epidemiological Study for Outcomes of Adjunctive Steroid Therapy in Pediatric Patients with Bacterial Meningitis in Taiwan"

_ijerph, 2021, doi:10.3390/ijerph18126386_

Round 1

Reviewer 1 Report

  1. The quality of English has to be significantly improved
  2. Paragraph 2.2, line 7. word missing after bacterial
  3. Figure 2 and Figure 3 contains mutually exclusive data. In figure 2 survival probability after 12 months is approximately 0.5 and 0.6 for steroid and placebo respectively vs 1,0 and 0,9 in Fig. 2.
  4. Data presented in the current paper has very limited value - both groups seem to differ significantly in terms of severity of disease course. Steroid group had significantly higher percentage of hydrocephalus, acute respiratory failure, brain oedema and brain abscesses. This suggests significantly more severe course in steroid group - therefore it seems that observed differences may be rather related to more severe course not to the steroid themselves. This was briefly raised in discussion. Nevertheless, a very important question emerges - what is the value of that data and what is the point of the whole article? Data does neither prove nor disprove hypothesis that steroid treatment lower survival rate.

Author Response

  1. The quality of English has to be significantly improved

Answers: Thanks for your suggestion. The manuscript had sent for English editing and A Native English Speaker edit the revised manuscript.

  1. Paragraph 2.2, line 7. word missing after bacterial

Answers: Thanks for your comment. We corrected the mistake

  1. Figure 2 and Figure 3 contains mutually exclusive data. In figure 2 survival probability after 12 months is approximately 0.5 and 0.6 for steroid and placebo respectively vs 1,0 and 0,9 in Fig. 2.

Answers: Thanks for your comments. We agree with your concern. Although we collected the data from NHIRD included one-year, three-year fatality and 5-year cases fatality between the two groups, the mean length of hospital stay (days) in both groups was also less than one month. After much discussion with all the co-authors, we decided to remove figure 2 because the figure was plotted incorrectly. I am sorry for the mistake. 

  1. Data presented in the current paper has very limited value - both groups seem to differ significantly in terms of severity of disease course. Steroid group had significantly higher percentage of hydrocephalus, acute respiratory failure, brain oedema and brain abscesses. This suggests significantly more severe course in steroid group-therefore it seems that observed differences may be rather related to more severe course not to the steroid themselves. This was briefly raised in discussion. Nevertheless, a very important question emerges-what is the value of that data and what is the point of the whole article? Data does neither prove nor disprove hypothesis that steroid treatment lower survival rate.

Answers: Thanks for your comments. Although there are several limitations, the role of adjunctive steroid therapy in pediatric bacterial meningitis in Taiwan did not demonstrate that can prevent neurologic complications or even reduce the mortality rate of bacterial meningitis.

Reviewer 2 Report

Thank you for the opportunity to review this manuscript.  I believe it addresses an important topic, and information in this area is lacking.  Overall, this is a well-written manuscript.  I believe the following comments/suggestions would strengthen the quality of the manuscript.

This is a retrospective study designed to look at individuals treated with corticosteroids and not treated with corticosteroids.  Using the term placebo group for those not treated makes it sound like the authors conducted an experimental trial and had a placebo treatment.  To be consistent with epidemiological observational designs (sounds like a retrospective cohort study), the groups should be referred to exposed and unexposed with exposed referring to those who received corticosteroid therapy.  The unexposed group is not a placebo, but rather a group that did not received the same exposure.  Also, be careful with wording such as “…were placed in the steroid and placebo groups…”  due to the observational nature of the design.

Please change “data was” to “data were”

There was a decrease in the number of cases in the exposed and unexposed groups from 2000 – 2013.  It would be helpful for the authors to elaborate further on this trend and comment on whether this would affect the severity of cases that were observed in the later years.

From the version I downloaded, it looks like the total number of cases in the unexposed group was cut off in Table 1.

Were the differences in causative pathogens listed in table 2 significantly different between groups?

It appears that the group exposed to corticosteroids was sicker.  Are steroids typically prescribed for more severe cases?  The authors briefly addressed this in the discussion.  If this is true, consider conducting a matched study where exposed and unexposed are matched based on severity or only include severe patients.  This would make the groups more comparable.

Author Response

Reviewer's comments:

  1. This is a retrospective study designed to look at individuals treated with corticosteroids and not treated with corticosteroids. Using the term placebo group for those not treated makes it sound like the authors conducted an experimental trial and had a placebo treatment. To be consistent with epidemiological observational designs (sounds like a retrospective cohort study), the groups should be referred to exposed and unexposed with exposed referring to those who received corticosteroid therapy.  The unexposed group is not a placebo, but rather a group that did not received the same exposure.  Also, be careful with wording such as “…were placed in the steroid and placebo groups…”  due to the observational nature of the design.

Answers: Thanks for your comments. Your concern is quite right. Using the term placebo in epidemiological observational study is inappropriate, and the terms exposed and unexposed groups are more proper. For easy understanding, we revised the manuscript and use the terms “steroid and non-steroid groups”.

  1. Please change “data was” to “data were”

Answers: We change “data was” into “data were” in accordance with your comment.

  1. There was a decrease in the number of cases in the exposed and unexposed groups from 2000 – 2013. It would be helpful for the authors to elaborate further on this trend and comment on whether this would affect the severity of cases that were observed in the later years.

Answers: Thanks for your comments. We updated discussion section explaining the possible reason of decreasing trend in number of bacterial meningitis, but we could not conclude whether this would affect the severity of cases.

In Discussion section, paragraph 4, we added the sentences as follows:

“In this study, there was a decreasing trend in the number of pediatric bacterial meningitis during the study period, and similar trend was also reported in prior study. We presumed that it was attributed to numerous factors, including fewer birth rate, better environmental sanitation, and vaccination administration.”

  1. From the version I downloaded, it looks like the total number of cases in the unexposed group was cut off in Table 1.

Answers: Thanks for your comments. It might be a typographic issue, we corrected in the revised version.

  1. Were the differences in causative pathogens listed in table 2 significantly different between groups?

Answers: Thanks for your comments. Indeed, it could be more informative to provide p value. It had better to update and enrichment with more current data beyond 2013 and make a statistical analysis between the causative pathogens. Because we need to make an appointment with the statistical center in the Ministry of Science and Technology of our country to obtain the new raw database (NHIRD), and the entire course needs one month to complete. The outbreak of the COVID-19 pandemic in Taiwan recently, and may affect our ability to make a new statistical analysis. However, the Editorial office did not respond to our request. We only should the percentage of each causative pathogen and did not make a statistical analysis.

  1. It appears that the group exposed to corticosteroids was sicker. Are steroids typically prescribed for more severe cases? The authors briefly addressed this in the discussion.  If this is true, consider conducting a matched study where exposed and unexposed are matched based on severity or only include severe patients.  This would make the groups more comparable.

Answers: Thanks for your wisdom to help the authors to have a better presentation of the manuscript. We agree with your comment that conducting a matched analysis based on the severity of two groups would make the results more convincing. As we mentioned above, we need more time to obtain the new raw database (NHIRD), and the outbreak of the COVID-19 pandemic in Taiwan recently and may affect our ability to make a new statistical analysis. I hope you can understand.

Reviewer 3 Report

Please note:  As line numbers are not supplied, I have copied and pasted the authorial statements I feel need editing and provided my comments in bold

Section 2.2

“We compared empirical antibiotic treatment in combination with systemic corticosteroid with empirical antibiotic treatment alone in patients with acute bacterial meningitis”

I recommend you define here that antibiotic treatment alone is the placebo group and steroid+treatment is the treatment group, possibly as such

We compared empirical antibiotic treatment in combination with systemic corticosteroid (treatment) with empirical antibiotic treatment alone (placebo) in patients with acute bacterial meningitis

Section 2.3

“Data were expressed as median (interquartile range [IQR])” – what data?  Continuous data?  This should be clarified.

Figure 2

The measurements in figure 2 are unclear.  Whatever event it is that this graph is measuring (the implication is that it is fatality but that should be clearly stated in caption, text, and on the axis label), appears to occur in 40 % of placebo group and 80% of steroid group.  These numbers do not make sense in the context of mortality (far too high).  If I as a researcher familiar with bacteria meningitis struggle to follow this graph due to lack of context, I imagine other readers will as well.  Please provide additional context.  Additionally, I only see mention of one-year fatality, so why are the curves still changing after 12 months? 

Section 3.1

“In total, 8083 episodes suffered from bacterial meningitis during the study period; of these, 26% (2122/8083) and 74% (5961/8083) were placed in the steroid and placebo groups, respectively. The 7561 patients included 4573 males (60.5%) and 2988 females (39.5%).”

Please round to a consistent number of digits-you present only whole numbers in the 1st sentence, then decimals in the second.

“The causative pathogens isolated from the CSF cultures of the 8083 episodes with bacterial meningitis are shown in Table 2. Streptococcus (S.) Pneumoniae was the most frequent one, accounting for 7.2% and 1.9% in the steroid group and placebo group, respectively, followed by Hemophilus (H.) influenza (4.0% and 0.6%) and Escherichia (E.) coli (0.7% and 0.3%).”

Please note that the organism is Haemophilus influenzae, not Hemophilus influenza

Table 1

You report mean for age.  I would like to see further stratification of this group.  Perhaps consider using the pediatric age stratification scheme suggested by Thigpen (Thigpen, Michael C., Cynthia G. Whitney, Nancy E. Messonnier, Elizabeth R. Zell, Ruth Lynfield, James L. Hadler, Lee H. Harrison et al. "Bacterial meningitis in the United States, 1998–2007." New England Journal of Medicine 364, no. 21 (2011): 2016-2025.)  Additionally, I took your comments in methods to suggest you were using median, please clarify

I very much appreciate the identification of species.  Too many manuscripts on bacterial meningitis fail to recognize the importance of species-level difference.  It would be an interesting addition, or perhaps follow up study, to conduct such curves on a species-stratified level. 

I find it highly unlikely you identified ~80% of species as the result of organisms other than S. pneumoniae, H. influenzae, or E. coli.  It seems more likely this is a missing classification.  Please review the data and if these cases are missing species, update this name.  If they are in fact other species, I recommend you shift the focus of the manuscript to explaining this unique epidemiology. 

Section 3.2

(11.0%) was the most common clinical feature, followed by acute respiratory failure (10.2%) and pneumonia (9.3%). In the placebo group, urinary tract infection (10.57%) was the most common clinical feature, followed by fever (6.7%) and pneumonia (4.0%).”

Please round consistently

Table 3.

While it is true you can make SAS give you a p value for a 0-1 comparison, the precision of this estimate would be weak.  I recommend you remove alcoholic liver cirrhosis.  Additionally, please provide confidence intervals for each of these calculations.  This is easily acquired from SAS and helps put the results in better context.

Section 3.3

The length of hospital stay (days) was 13 (range: 6–27) and 6 (range: 4–10) days respectively in the steroid and placebo groups”

This should say, the median length of stay

Discussion

The other study about a meta-analysis of corticosteroids for acute bacterial meningitis showed a reduction of mortality rate in S. pneumoniae meningitis under corticosteroid treatment”

I find this very misleading.  The actual manuscript you reference states specifically that “Corticosteroids significantly reduced hearing loss and neurological sequelae, but did not reduce overall mortality”  It is true that a subgroup analysis of this study found the result you describe in S pneumo, but to not mention that all other subgroups found no effect is misleading.  Since you examined bacterial meningitis without subgrouping, it seems your discussion should mention the non-subgrouped result, as it is most comparable to your work, and that states the opposite of what you have summarized

Pneumococcal vaccination program for children aged 2 to 5 was started in 2013 in Taiwan, and further surveillance data to identify the effectiveness in the prevention of S. pneumoniae meningitis was warranted”

This should read “A pneumococcal vaccination program…”

“The corticosteroid therapy efficacy of reducing hearing loss was shown in children with Hib meningitis, but this efficacy was not demonstrated in children with meningitis caused by non-Haemophilus species”

Italicize Haemophilus

In another retrospective observational study, regardless of the steroid or placebo group, brain abscess was identified as the most common neurologic complication in adult bacterial meningitis in Taiwan, but only a small part of patients presented with hydrocephalus and brain edema[22]”

I recommend changing part to portion here

Except for H. influenzae (ICD-9-CM codes 3200), S. pneumoniae (ICD-9-CM codes 3201), and Escherichia coli (ICD-9-CM codes 320.82), not all the causative pathogens had ICD-9-CM codes numbers could not be recorded in the NHIRD nationwide population-based database. Therefore, we could not analyze the outcomes of adjunctive steroid therapy in individualized causative pathogens.”

I recently conducted a study with a similar design (ICD code harvesting), though I was examining different variables.  It was my experience that ~70% of cases coded bacterial meningitis but lacking a code for H influenzae, S pnuemo, or E coli, were actually caused by those agents but simply not coded.  I am unfamiliar with how things are done in the Taiwanese health system, but it is possible these were in fact the causal agents but that only a broader code for bacterial meningitis was applied as was the case in my study.  I will say you still have the sample size to at least do so in S pneumo and probably in H influenzae and Gram neg rods in general.

Therefore, the role of the adjunctive corticosteroid therapy was hard to evaluate, and a statistical bias existed in the study.

This is well said.

The beneficial effects of the adjunctive corticosteroid therapy in pediatric bacterial meningitis remain inconclusive, we look forward to more prospective multicenter investigations to clarify the role of adjunctive corticosteroid therapy in pediatric bacterial meningitis in Taiwan.

I agree with this conclusion

Author Response

Reviewer's comments:

  1. Section 2.2

“We compared empirical antibiotic treatment in combination with systemic corticosteroid with empirical antibiotic treatment alone in patients with acute bacterial meningitis”

I recommend you define here that antibiotic treatment alone is the placebo group and steroid + treatment is the treatment group, possibly as such We compared empirical antibiotic treatment in combination with systemic corticosteroid (treatment) with empirical antibiotic treatment alone (placebo) in patients with acute bacterial meningitis

Answers: Thanks for your comments. We use the steroid group and non-steroid group throughout the manuscript to make it clearer.

In section 2.2, paragraph 2 of the revised version. We advised the sentences as follows:

“We compared empirical antibiotic treatment in combination with systemic cortico-steroids (steroid group) with empirical antibiotic treatment alone (non-steroid group) in patients with acute bacterial meningitis.”

  1. Section 2.3

“Data were expressed as median (interquartile range [IQR])” – what data?  Continuous data?  This should be clarified.

Answers: In Section 2.3, we revised this paragraph to make it clearer. They are as follows:

“Data of length of hospital stay (days) and medical costs of hospitalization (New Taiwan dollars, NTD) are expressed as medians (interquartile range [IQR]).”

  1. Figure 2

The measurements in figure 2 are unclear.  Whatever event it is that this graph is measuring (the implication is that it is fatality but that should be clearly stated in caption, text, and on the axis label), appears to occur in 40 % of placebo group and 80% of steroid group.  These numbers do not make sense in the context of mortality (far too high).  If I as a researcher familiar with bacteria meningitis struggle to follow this graph due to lack of context, I imagine other readers will as well.  Please provide additional context.  Additionally, I only see mention of one-year fatality, so why are the curves still changing after 12 months?

Answers: Thanks for your comments. We agree with your concern. Although we collected the data from NHIRD included one-year, three-year fatality and 5-year cases fatality between two groups, the mean length of hospital stay (days) in both groups were also less than one month. After much discussion with all the co-authors, we decided to remove figure 2 because the figure was plotted incorrectly. I am sorry for the mistake. 

  1. Section 3.1

“In total, 8083 episodes suffered from bacterial meningitis during the study period; of these, 26% (2122/8083) and 74% (5961/8083) were placed in the steroid and placebo groups, respectively. The 7561 patients included 4573 males (60.5%) and 2988 females (39.5%).”

Please round to a consistent number of digits-you present only whole numbers in the 1st sentence, then decimals in the second.

Answers: Thanks for your comments. We correct corrected it in revised version.

  1. Please note that the organism is Haemophilus influenzae, not Hemophilus influenza

Answers: Thanks for your comments. We correct corrected the mistake in revised version.

  1. Table 1

You report mean for age.  I would like to see further stratification of this group.  Perhaps consider using the pediatric age stratification scheme suggested by Thigpen (Thigpen, Michael C., Cynthia G. Whitney, Nancy E. Messonnier, Elizabeth R. Zell, Ruth Lynfield, James L. Hadler, Lee H. Harrison et al. "Bacterial meningitis in the United States, 1998–2007." New England Journal of Medicine 364, no. 21 (2011): 2016-2025.)  Additionally, I took your comments in methods to suggest you were using median, please clarify

Answers: Thanks for your comments. Indeed, it could be more informative to use the pediatric age stratification scheme suggested by Thigpen and update and enrichment with more current data beyond 2013. Because we need to make an appointment with the statistical center in the Ministry of Science and Technology of our country to obtain the new raw database (NHIRD), and the entire course needs one month to complete. The outbreak of the COVID-19 pandemic in Taiwan recently, and may affect our ability to make a new statistical analysis. However, the Editorial office did not respond to our request. We can only keep the present presentation form and hope you can understand.

Furthermore, in the statistical analysis, we revised the sentences as follows:

Data are expressed as the mean ± standard deviation or median (interquartile range). Age at infection is expressed as mean ± standard deviation. Data of length of hospital stay (days) and medical costs of hospitalization (New Taiwan dollars, NTD) are expressed as median (interquartile range [IQR]).

  1. I very much appreciate the identification of species. Too many manuscripts on bacterial meningitis fail to recognize the importance of species-level difference. It would be an interesting addition, or perhaps follow up study, to conduct such curves on a species-stratified level.

Answers: We agreed with your comment that many papers on bacterial meningitis fail to recognize the importance of species-level difference. Currently because of the limited data, please allow authors to keep the study design and look forward to species-stratified level analysis in the future.

  1. I find it highly unlikely you identified ~80% of species as the result of organisms other than S. pneumoniae, H. influenzae, or E. coli. It seems more likely this is a missing classification. Please review the data and if these cases are missing species, update this name. If they are in fact other species, I recommend you shift the focus of the manuscript to explaining this unique epidemiology.

Answers: Thanks for your comments. Due to the initial study design and limited information in raw data, we could not re-check the causative pathogens in current data. The aim of this study is to explore the outcome of adjunctive steroid therapy in pediatric meningitis. We updated part of discussion section about causative pathogen, but did not have further explanation about this epidemiology. Please allow authors to keep this revised version.

  1. Section 3.2

“(11.0%) was the most common clinical feature, followed by acute respiratory failure (10.2%) and pneumonia (9.3%). In the placebo group, urinary tract infection (10.57%) was the most common clinical feature, followed by fever (6.7%) and pneumonia (4.0%).”

Please round consistently

Answers:Thanks for your kind reminder, we corrected it in the revised version.

  1. Table 3.

While it is true you can make SAS give you a p value for a 0-1 comparison, the precision of this estimate would be weak.  I recommend you remove alcoholic liver cirrhosis.  Additionally, please provide confidence intervals for each of these calculations.  This is easily acquired from SAS and helps put the results in better context.

Answers: We agree with your comment that about 70% of cases coded bacterial meningitis but lacking a code for H influenzae, S pneumoniae, or E coli, were caused by those agents but simply not coded from not be recorded in the NHIRD nationwide population-based database. It is a retrospective study. If we had more time to revise the manuscript, we can conduct a matched analysis based on the severity of the two groups could make the results more convincing.

  1. Section 3.3

“The length of hospital stay (days) was 13 (range: 6–27) and 6 (range: 4–10) days respectively in the steroid and placebo groups”

This should say, the median length of stay

Answers:Thanks for your comments. We corrected it in the revised version.

  1. Discussion

“The other study about a meta-analysis of corticosteroids for acute bacterial meningitis showed a reduction of mortality rate in S. pneumoniae meningitis under corticosteroid treatment” I find this very misleading. The actual manuscript you reference states specifically that “Corticosteroids significantly reduced hearing loss and neurological sequelae, but did not reduce overall mortality” It is true that a subgroup analysis of this study found the result you describe in S pneumoniae, but to not mention that all other subgroups found no effect is misleading. Since you examined bacterial meningitis without subgrouping, it seems your discussion should mention the non-sub grouped result, as it is most comparable to your work, and that states the opposite of what you have summarized

Answers: Thanks for your comments. We agree with your comment that this reference showed insufficient evidence that corticosteroids reduce overall mortality, and only the subgroup analyses showed mortality reduction for S. pneumoniae meningitis. To make it clearer, we revised the sentences in Discussion section, paragraph 1 as follows.

“A meta-analysis of the use of corticosteroids for acute bacterial meningitis showed no significant reduction in overall mortality, but demonstrated a reduction in mortality rate for S. pneumoniae meningitis in subgroup analyses for causative organisms.” and

“Due to limited data, this study did not perform subgroup analyses of reduction in mortality for causative organisms.”

  1. “Pneumococcal vaccination program for children aged 2 to 5 was started in 2013 in Taiwan, and further surveillance data to identify the effectiveness in the prevention of S. pneumoniae meningitis was warranted”

This should read “A pneumococcal vaccination program…”

Answers: Thanks for your comments. We corrected it in the revised version.

  1. “The corticosteroid therapy efficacy of reducing hearing loss was shown in children with Hib meningitis, but this efficacy was not demonstrated in children with meningitis caused by non-Haemophilus species”. Italicize Haemophilus

Answers: Thanks for your comments. We corrected the mistake.

  1. “In another retrospective observational study, regardless of the steroid or placebo group, brain abscess was identified as the most common neurologic complication in adult bacterial meningitis in Taiwan, but only a small part of patients presented with hydrocephalus and brain edema [22]”

I recommend changing part to portion here

Answers: Thanks for your comments. We corrected it in the revised version.

  1. “Except for H. influenzae (ICD-9-CM codes 3200), S. pneumoniae (ICD-9-CM codes 3201), and Escherichia coli (ICD-9-CM codes 320.82), not all the causative pathogens had ICD-9-CM codes numbers could not be recorded in the NHIRD nationwide population-based database. Therefore, we could not analyze the outcomes of adjunctive steroid therapy in individualized causative pathogens.” I recently conducted a study with a similar design (ICD code harvesting), though I was examining different variables. It was my experience that ~70% of cases coded bacterial meningitis but lacking a code for H influenzae, S pneumoniae, or E coli, were caused by those agents but simply not coded. I am unfamiliar with how things are done in the Taiwanese health system, but it is possible these were in fact the causal agents but that only a broader code for bacterial meningitis was applied as was the case in my study.  I will say you still have the sample size to at least do so in S pneumoniae and probably in H influenzae and Gram neg rods in general.

Answers: We agree with your comment that about 70% of cases coded bacterial meningitis but lacking a code for H influenzae, S pneumoniae, or E coli, were caused by those agents but simply not coded from not be recorded in the NHIRD nationwide population-based database. It is a retrospective study. If we had more time to revise the manuscript, we can conduct a matched analysis based on the severity of the two groups could make the results more convincing.

Reviewer 4 Report

This is a retrospective cohort study based on a national database, evaluating the benefit of systemic corticosteroid therapy in the setting of bacterial meningitis. As the author reported, patients treated with systemic steroid had worse outcomes which is likely due to severity of their illness and not the intervention. The study has one major limitation as noted in the discussion with majority of patients’ pathogen not identified and reported as “others”.

It should also be noted that prior studies have showed that optimal response to corticosteroid therapy occurred when treatment is administered prior to antibiotic therapy or at time of antibiotic therapy. However, the authors do not report the time between corticosteroid therapy and antibiotic therapy. The authors should include this in their analysis, or they should update the limitation section.

Additional limitations

  • To facilitate readability of the abstract and the manuscript, the authors should be consistent when reporting results and use the same patient group sequence. Following the tables, it would be better to report Steroid group first followed by placebo group.
  • Tables 1, 5 and 6. Specify unit used for age variable
  • Section 3.4, tables 5 and 6. Specify the denominator for case fatality rates
  • Discussion (page 7, end of 2nd paragraph). It appears that pneumococcal vaccine was available to certain patients prior to 2013. The current study (table 1) demonstrates a decrease in number of bacterial meningitis which is likely secondary to pneumococcus and HiB vaccination. The authors should evaluate the trend of different causative agents based on year to help determine the effect of pneumococcal vaccine.
    • https://www.cdc.gov.tw/En/Category/ListContent/bg0g_VU_Ysrgkes_KRUDgQ?uaid=lS42udX_s0u2fN0qLcdrnw#:~:text=Two%20different%20pneumococcal%20vaccines%20are,vaccines%20(or%20killed%20vaccines).
  • As patients with tuberculous meningitis have improved outcomes when treated with corticosteroid, the authors should identify if any patients were diagnosed with tuberculous meningitis.
    • Prasad K, Singh MB, Ryan H. Corticosteroids for managing tuberculous meningitis. Cochrane Database Syst Rev. 2016 Apr 28;4(4):CD002244. doi: 10.1002/14651858.CD002244.pub4. PMID: 27121755; PMCID: PMC4916936.
  • The authors should specify if patients had indwelling ventricular catheters to differentiate between patients that had bacterial meningitis due to community pathogen versus patients with healthcare associated ventriculitis and meningitis.
  • As females appeared to be an independent variable associated with increased mortality, the authors should elaborate on this difference in the discussion section
  • Prior studies showed improved long-term sequelae with corticosteroid therapy, especially with bacterial meningitis secondary to Haemophilus influenzae. If available, the authors should include if patients had any sensorineural hearing loss or neurological deficits secondary to meningitis and if rates differed between the two groups. If the data is not available, this should be as a limitation in the discussion section

Author Response

Reviewer's comments:

  1. 1. This is a retrospective cohort study based on a national database, evaluating the benefit of systemic corticosteroid therapy in the setting of bacterial meningitis. As the author reported, patients treated with systemic steroid had worse outcomes which is likely due to severity of their illness and not the intervention. The study has one major limitation as noted in the discussion with majority of patients’ pathogen not identified and reported as “others”.

It should also be noted that prior studies have showed that optimal response to corticosteroid therapy occurred when treatment is administered prior to antibiotic therapy or at time of antibiotic therapy. However, the authors do not report the time between corticosteroid therapy and antibiotic therapy. The authors should include this in their analysis, or they should update the limitation section.

Answers: Thanks for your comments. This study was designed to explore the outcomes of adjunctive steroid therapy in pediatric bacterial meningitis. However, many limitations were encountered due to limited data recorded in NHIRD. We updated the limitation section in the revised version. They are as follows:

“Besides, the timing of administering dexamethasone was suggested prior to or at the same time as the first dose of antimicrobial therapy and might be less beneficial if late administration. Due to the nature of database analysis, our data did not include the timing between administration of corticosteroids and antibiotics.”

Additional limitations

  1. To facilitate readability of the abstract and the manuscript, the authors should be consistent when reporting results and use the same patient group sequence. Following the tables, it would be better to report Steroid group first followed by placebo group.

Answers: Thanks for your comments to make our manuscript more readable. We addressed this issue during revision.

  1. Tables 1, 5 and 6. Specify unit used for age variable

Answers: Thanks for your comments. We added the unit for age variable in the revised version.

  1. Section 3.4, tables 5 and 6. Specify the denominator for case fatality rates

Answers: Thanks for your comments. We updated section 3.4 and specified the denominator for case fatality rates. In order to keep the tables clear, please allow authors to keep current presentation of tables 5 and 6.

  1. Discussion (page 7, end of 2nd paragraph). It appears that pneumococcal vaccine was available to certain patients prior to 2013. The current study (table 1) demonstrates a decrease in number of bacterial meningitis which is likely secondary to pneumococcus and HiB vaccination. The authors should evaluate the trend of different causative agents based on year to help determine the effect of pneumococcal vaccine.

Answers: Thanks for your comments. We updated discussion section explaining the possible reason of decreasing trend in number of bacterial meningitis. Using current data, we could not evaluate each causative pathogen based on year.

In discussion section, paragraph 4, we revised the sentences as follows:

“In this study, there was a decreasing trend in the number of pediatric bacterial meningitis during the study period, and similar trend was also reported in prior study. We presumed that it was attributed to numerous factors, including fewer birth rate, better environmental sanitation, and vaccination administration.”

  1. As patients with tuberculous meningitis have improved outcomes when treated with corticosteroid, the authors should identify if any patients were diagnosed with tuberculous meningitis.

Answers: Thanks for your comments. Our study obtained ICD-9 codes for bacterial meningitis and excluded tuberculous meningitis.

  1. The authors should specify if patients had indwelling ventricular catheters to differentiate between patients that had bacterial meningitis due to community pathogen versus patients with healthcare associated ventriculitis and meningitis.

Reply from authors:

Thanks for your comments. It might be interesting to perform subgroup analyses, such as sex, age, causative pathogens, or route for infection. However, our current data did not include such detailed information of each patient for further subgroup analyses. Please allow authors to keep the study design.

  1. As females appeared to be an independent variable associated with increased mortality, the authors should elaborate on this difference in the discussion section

Reply from authors:

Thanks for your comments. Several studies have been devoted to sex differences in health and mortality. For the research on sex differences for bacterial meningitis, most studies had reported the clinical features, causative pathogens, treatment, and prognostic factors in adults with bacterial meningitis. However, there seems be no such discussion focused on sex differences of pediatric meningitis. We updated discussion section in revised version.

Please look into the revised version, discussion section, paragraph 2:

“In our study, females had significantly higher in-hospital fatality and 1-year fatality. Sex differences in health and mortality have been of longstanding interest to researchers, and several studies have been devoted to this topic. One prospective analysis investigating sex difference in adults with community-acquired bacterial meningitis showed that male sex was an independent predictor of unfavorable outcome and death. In contrast, another retrospective descriptive study reported females with urgent treatable etiologies to be a predictive factor of poor outcome. For pediatric bacterial meningitis, the studies to explore sex-based differences in mortality are scarce and have inconsistent results. The reason of discrepancy in mortality between males and females was uncertain, might be due to the differences of underlying development processes, immune and endocrine system.”

  1. Prior studies showed improved long-term sequelae with corticosteroid therapy, especially with bacterial meningitis secondary to Haemophilus influenzae. If available, the authors should include if patients had any sensorineural hearing loss or neurological deficits secondary to meningitis and if rates differed between the two groups. If the data is not available, this should be as a limitation in the discussion section

Reply from authors:

Thanks for your comments. Yes, the reviewer is right, investigating improvement of long-term sequelae with adjunctive corticosteroid therapy is important, especially in certain causative pathogens. However, due to the observational nature of this study, some clinical information was not completely recorded in the database. It would be useful to have more data on clinical manifestation. We updated limitation in revised version.

Please look into the revised version, discussion section, the second to last paragraph:

“Finally, it is important to explore if improvement of long-term sequelae with adjunctive corticosteroid therapy, especially in certain causative pathogens, but this study did not include such analysis due to limited data. It would be useful to have more data on the clinical manifestation of each patient to better evaluate neurologic outcome after adjunctive corticosteroid therapy.”

Reviewer 5 Report

Title of article reviewed:  Nationwide population-based epidemiologic study for outcomes of adjunctive steroid therapy in pediatric patients with bacterial meningitis in Taiwan

GENERAL COMMENTS: The present work, by Hsien et al. presents the results of a comparable study in a pediatric population in Taiwan, in order to define potential beneficial effects of the adjunctive use of corticosteroids in bacterial meningitis in children.

The study seems interesting as it is based in the analysis of a large amount of data obtained from a long-term 14-year period. However, the data used are from the years 2000-2013 and there is a need for update and enrichment with more current data beyond 2013, since the authors can benefit from the access and use of the large digital data base (NHIRD). Furthermore, references used must also be updated, as more recent and more relative references could be found  (e.g. Wang et al. 2018, Hasbun 2019). The discussion section needs to be more analytical, related to the results. Finally, English language must be improved throughout the whole manuscript.

As the whole manuscript needs major revision, the authors are kindly advised to specifically consider the following:  

SPECIFIC COMMENTS

Title: The word “epidemiologic” needs to be replaced by “epidemiological”

Patients and Methods:

In the text, the authors use the words “patients”, “cases” and “episodes”. The authors are kindly advised to clarify the definitions and resume the data, since this causes confusion to the reader.

Furthermore, the authors are kindly advised to clarify and define the two formed groups “placebo” and “steroid” and the criteria by which these groups were formed.

Results

May the authors consider presenting Figures 2 and 3, since they are neither mentioned, nor explained in the results session.

In paragraph 3.4, the study for the one-year follow up needs further clarification. For example, which patients were followed up and for what reason (e.g. due to underlying conditions?).

The authors are kindly requested to check again the sum of causative agents in Table 2, as the value seems to be rather confusing and does not add up to the total number of the study population. Furthermore, in tables 5 and 6 authors may consider defining the patients with underlying disease with “survival” or “death” outcome and discuss potential effect of underlying disease on the outcome.

The authors are kindly requested to check the spelling of all microorganisms’ genus and species names, which must be written in italics format.

Author Response

Reviewer's comments:

  1. The study seems interesting as it is based in the analysis of a large amount of data obtained from a long-term 14-year period. However, the data used are from the years 2000-2013 and there is a need for update and enrichment with more current data beyond 2013, since the authors can benefit from the access and use of the large digital data base (NHIRD).

Reply from authors:

Thanks for your comments. These data had been obtained and this study had been designed years before, but we did not complete this manuscript until this year. Indeed, an update and enrichment with more current data beyond 2013 will make this manuscript more interesting. However, we have few days to revise this manuscript, and the entire course to obtain new raw database from NHIRD is really time consuming. Please allow authors to keep current study design.

  1. Furthermore, references used must also be updated, as more recent and more relative references could be found (e.g. Wang et al. 2018, Hasbun 2019).

Answers: Thanks for your comments. We update more recent and more relative references as your request.

  1. The discussion section needs to be more analytical, related to the results.

Answers: Thanks for your comments. In the Discussion section, we try to make it more analytical, related to the results as your suggestion. However, many limitations were encountered due to limited data recorded in NHIRD. We updated the discussion section and the limitation section in the revised version.

  1. Finally, English language must be improved throughout the whole manuscript.

Answers: Thanks for your suggestion. The manuscript had sent for English editing and A Native English Speaker edit the revised manuscript.

SPECIFIC COMMENTS

  1. Title: The word “epidemiologic” needs to be replaced by “epidemiological”

Answers: Thanks for your comments. We change “epidemiologic” into “epidemiological” in accordance with your comments.

  1. Patients and Methods:

In the text, the authors use the words “patients”, “cases” and “episodes”. The authors are kindly advised to clarify the definitions and resume the data, since this causes confusion to the reader.

Answers: Thanks for your comments. We thought that “patients” is equivalent to “cases” and we use them interchangeably. “Episode” means one episode of meningitis.

To make it clearer, we revised the sentences in section 3.1. They are as follows:

“There were 7561 cases of bacterial meningitis, of which 522 cases represented recurrent episodes. In total, there were 8083 episodes of bacterial meningitis during the study period. Of these, 26.3% (2122/8083) and 73.7% (5961/8083) were placed in the steroid and non-steroid groups, respectively.”

  1. Furthermore, the authors are kindly advised to clarify and define the two formed groups “placebo” and “steroid” and the criteria by which these groups were formed.

Reply from authors:

Thanks for your suggestion to make the manuscript to be more readable. For this study with epidemiological observational design, we will use the terms “steroid and non-steroid groups” instead of “steroid and placebo groups” in this manuscript.

In section 2.2, paragraph 2, we revised the sentences as follows:

“We compared empirical antibiotic treatment in combination with systemic cortico-steroids (steroid group) with empirical antibiotic treatment alone (non-steroid group) in patients with acute bacterial meningitis.”

  1. Results

May the authors consider presenting Figures 2 and 3, since they are neither mentioned, nor explained in the results session.

Answers: Thanks for your comments. , the mean length of hospital stay (days) in both groups were also less than one month. After much discussion with all the co-authors, we decided to remove figure 2 because the figure was plotted incorrectly. I am sorry for the mistake.  We added the following sentences in Results section, and they are as follows:

To look at each factor individually, we calculated Kalpan-Meier estimates of the fraction developing survival probability (survival and non-survival outcome) in patients with acute bacterial meningitis between two groups (steroid and non-steroid groups) during the 1-year follow-up period and tested for differences by using a log-rank test. The results showed statistical significance (P<0.0001) (Figure 2)

  1. In paragraph 3.4, the study for the one-year follows up needs further clarification. For example, which patients were followed up and for what reason (e.g. due to underlying conditions?).

Answers: One year follow-up was according to database from recorded in NHIRD. Several patients follow up at outpatient clinic due the neurological sequelae after bacterial meningitis. We can only find the patients were survival or not

  1. The authors are kindly requested to check again the sum of causative agents in Table 2, as the value seems to be rather confusing and does not add up to the total number of the study population. Furthermore, in tables 5 and 6 authors may consider defining the patients with underlying disease with “survival” or “death” outcome and discuss potential effect of underlying disease on the outcome.

Answers: Thanks for your comments. We reviewed the data and corrected the error in Table 2. To make it clearer, the mixed infections were wrongly added to others.

Patient’s conditions for fatality or not was according to database from recorded in NHIRD and potential effect of underlying disease on the outcome could be difficult to analysis due to the limitation from database in NHIRD.

  1. The authors are kindly requested to check the spelling of all microorganisms’ genus and species names, which must be written in italics format.

Answers: Thanks for your comments. We corrected it after English revision.

Round 2

Reviewer 1 Report

Authors have sufficiently addressed all my previous comments.

Author Response

The authors have sufficiently addressed all my previous comments.

Answers: Thanks for your comment

Reviewer 2 Report

Thank you for sending the revised manuscript.  The changes have really improved the overall quality of the manuscript.

Author Response

Thank you for sending the revised manuscript.  The changes have really improved the overall quality of the manuscript.

Answers: Thanks for your comment

Reviewer 4 Report

Thank you for updating the manuscript. The authors have addressed the previous comments and present a complete study. 

The only limitation not addressed is 

  1. As patients with tuberculous meningitis have improved outcomes when treated with corticosteroid, the authors should identify if any patients were diagnosed with tuberculous meningitis.

Please update method or discussion section to clarify that patients with tuberculous meningitis were excluded from the study. 

Author Response

Thank you for updating the manuscript. The authors have addressed the previous comments and present a complete study. The only limitation not addressed is As patients with tuberculous meningitis have improved outcomes when treated with corticosteroid, the authors should identify if any patients were diagnosed with tuberculous meningitis. Please update method or discussion section to clarify that patients with tuberculous meningitis were excluded from the study.

Answers: Thanks for your comment. We added the following sentence in 2.2. Study Subjects and they are as follows:

 Patients with tuberculous meningitis were excluded from this study.

Reviewer 5 Report

Title of article reviewed:  Nationwide population-based epidemiological study for outcomes of adjunctive steroid therapy in pediatric patients with bacterial meningitis in Taiwan

GENERAL COMMENTS:

The present work, by Hsien et al. presents the results of a comparable study in a pediatric population in Taiwan, in order to define potential beneficial effects of the adjunctive use of corticosteroids in bacterial meningitis in children.

This is the second review of the above article presenting an interesting study from a long-term 14-year period. Authors can keep the present design, as they consider that adding more data would be more time-consuming.

Authors have made a great work in revising the manuscript. English language is greatly improved throughout the whole manuscript and it is easier for the reader to follow the flow. As well, they have successfully define “cases” and “episodes” and they have successfully work on the revising of the discussion session.  

SPECIFIC COMMENTS

Patients and Methods:

“Cases” and “episodes”. are very well defined in section 3.1.  Authors are kindly advised to follow that definition throughout the whole manuscript (e.g. abstract, line 25, Figure 1). Please check the numbers throughout the whole manuscript so to be easy to define if it is referred to “cases” or “episodes” according to the definition of section 3.1.

 “Placebo” group has been successfully replaced by “non-steroid” group.

 In section 2.2 maybe authors should clarify that cases were selected from the database and criteria for the 2 groups are not part of the study, as drug administration has been already driven by clinicians.

It is suggested that it may would be more clear, if the sentences in lines 126-129 with the cases obtained from the database “There were….respectively”, as well as Figure 1, were transferred in section 2.2.

Results

Τhe authors correctly improved paragraph 3.4. They are kindly requested to further analyze the sentence in line 169: «the results showed statistical significance”. Also, they are advised to clearly refer the limitation of the database in access of the potential effect of the underlying disease. Is it also considered a limitation of the method? Is it something of great significance that would influence the results? Maybe authors have to consider discussing about it in the discussion section, or add it to the limitations of the method, also they can refer if other studies have worked with the influence of the underlying disease on the outcome.

In Table 2. the reference in mixed infections is rather confusing than useful. Did all the mixed infections were due to microorganisms other than the microorganisms that are already in the table?-if yes, they are already included in “Others”. If no, maybe the authors have to consider additional results: e.g. if S. pneumoniae  is included in one mixed infection together with another microorganism in the steroid group, the total should be 153.

Finally, the discussion session was greatly improved with additional paragraphs, although authors are kindly advised to rearrange the paragraphs in order to have a better flow. I would suggested to start with the general info, then continue with the method, the results and the limitations and finally correlate them with those of previous studies. References can be a little bit more improved (there are relative references available of the two last years).

Author Response

Patients and Methods:

  1. “Cases” and “episodes”. are very well defined in section 3.1. Authors are kindly advised to follow that definition throughout the whole manuscript (e.g. abstract, line 25, Figure 1). Please check the numbers throughout the whole manuscript so to be easy to define if it is referred to “cases” or “episodes” according to the definition of section 3.1.

Answers: Thanks for your comments. We updated abstract section and Figure 1. “Placebo” group has been successfully replaced by “non-steroid” group.

  1. In section 2.2 maybe authors should clarify that cases were selected from the database and criteria for the 2 groups are not part of the study, as drug administration has been already driven by clinicians.

Answers: In order to clarify those cases that were selected from the database. We add the following sentence in 2.2. Study Subjects and Figure 1 as follows:

  There were 7561 cases of bacterial meningitis, of which 522 cases represented recurrent episodes. In total, there were 8083 episodes of bacterial meningitis during the study period. Of these, 26.3% (2122/8083) and 73.7% (5961/8083) were placed in the steroid and non-steroid groups, respectively.

  1. It is suggested that it may would be more clear, if the sentences in lines 126-129 with the cases obtained from the database “There were….respectively”, as well as Figure 1, were transferred in section 2.2.

Answers: Again, in order to clarify those cases that were selected from the database. We add the following sentence in 2.2. Study Subjects and Figure 1 as follows:

  There were 7561 cases of bacterial meningitis, of which 522 cases represented recurrent episodes. In total, there were 8083 episodes of bacterial meningitis during the study period. Of these, 26.3% (2122/8083) and 73.7% (5961/8083) were placed in the steroid and non-steroid groups, respectively.

  1. The authors correctly improved paragraph 3.4. They are kindly requested to further analyze the sentence in line 169: «the results showed statistical significance”. Also, they are advised to clearly refer the limitation of the database in access of the potential effect of the underlying disease. Is it also considered a limitation of the method? Is it something of great significance that would influence the results? Maybe authors have to consider discussing about it in the discussion section, or add it to the limitations of the method, also they can refer if other studies have worked with the influence of the underlying disease on the outcome.

Answers: Thanks for your comments. Your concern quite right, the two patient groups might have different underlying disease and disease severity. Conducting a matched analysis based on severity or different underlying disease of two groups would make the results more convincing. We would add it to the limitations. We add the following sentences in Discussion as the limitation of the study. They are as follows:

Third, the patients in two groups might have different disease severity and underlying disease. Conducting a matched analysis based on severity or different underlying disease of two groups would make the results more convincing, and further studies might be needed to clarify the influence of the underlying disease on the outcome.

  1. In Table 2. the reference in mixed infections is rather confusing than useful. Did all the mixed infections were due to microorganisms other than the microorganisms that are already in the table?-if yes, they are already included in “Others”. If no, maybe the authors have to consider additional results: e.g. if S. pneumoniae is included in one mixed infection together with another microorganism in the steroid group, the total should be 153.

Answers: In our study, patients were considered to have mixed infection if at least two bacterial organisms were isolated concomitantly from the initial CSF cultures. The definition was according to our previous study. (Chang WN, Lu CH, Huang CR, Chuang YC. Mixed infection in adult bacterial meningitis. Infection 2000;28:8‑12) Almost those patients were in both nosocomial and postneurosurgical state. The purpose of this study is to explore the role of adjunctive steroid therapy in outcome of pediatric meningitis. To make it clear, we include those cases with mixed infection of bacterial meningitis into "Others" pathogens in Table 2 without changing the integrality of the manuscript.

  1. Finally, the discussion session was greatly improved with additional paragraphs, although authors are kindly advised to rearrange the paragraphs in order to have a better flow. I would suggest to start with the general info, then continue with the method, the results and the limitations and finally correlate them with those of previous studies. References can be a little bit more improved (there are relative references available of the two last years).

Answers: Thanks for your comment. We revised the discussion section in a concise way in accordance with your comment and several references published in the two last years. We added one meta-analysis on the role of adjunctive dexamethasone therapy in pediatric bacterial meningitis in Discussion section, and they are as follows:

One recent meta-analysis investigating the role of adjunctive dexamethasone therapy in pediatric bacterial meningitis demonstrated that dexamethasone had no significant effect on the follow-up mortality [29].

One recent meta-analysis investigating the role of adjunctive dexamethasone therapy in pediatric bacterial meningitis demonstrated that dexamethasone had no significant effect on the follow-up mortality [29].

  1. Wang Y, Liu X, Wang Y, Liu Q, Kong C, Xu G. Meta-analysis of adjunctive dexamethasone to improve clinical outcome of bacterial meningitis in children. Childs Nerv Syst. 2018;34(2):217-223.
  2. Alamarat Z, Hasbun R. Management of Acute Bacterial Meningitis in Children. Infect Drug Resist. 2020;13:4077-4089.
  3. Dunbar M, Shah H, Shinde S, et al. Stroke in Pediatric Bacterial Meningitis: Population-Based Epidemiology. Pediatr Neurol. 2018;89:11-18.